# Molecular Network Profiling in Intestinal- and Diffuse-Type Gastric Cancer

**DOI:** 10.3390/cancers12123833

**Published:** 2020-12-18

**Authors:** Shihori Tanabe, Sabina Quader, Ryuichi Ono, Horacio Cabral, Kazuhiko Aoyagi, Akihiko Hirose, Hiroshi Yokozaki, Hiroki Sasaki

**Affiliations:** 1Division of Risk Assessment, Center for Biological Safety and Research, National Institute of Health Sciences, Kawasaki 210-9501, Japan; hirose@nihs.go.jp; 2Innovation Center of NanoMedicine (iCONM), Kawasaki Institute of Industrial Promotion, Kawasaki 210-0821, Japan; sabina-q@kawasaki-net.ne.jp; 3Division of Cellular and Molecular Toxicology, Center for Biological Safety and Research, National Institute of Health Sciences, Kawasaki 210-9501, Japan; onoryu@nihs.go.jp; 4Department of Bioengineering, Graduate School of Engineering, University of Tokyo, Tokyo 113-0033, Japan; horacio@bmw.t.u-tokyo.ac.jp; 5Department of Clinical Genomics, National Cancer Center Research Institute, Tokyo 104-0045, Japan; kaaoyagi@ncc.go.jp; 6Department of Pathology, Kobe University of Graduate School of Medicine, Kobe 650-0017, Japan; hyoko@med.kobe-u.ac.jp; 7Department of Translational Oncology, National Cancer Center Research Institute, Tokyo 104-0045, Japan; hksasaki@ncc.go.jp

**Keywords:** cancer stem cell, epithelial-mesenchymal transition, molecular network

## Abstract

**Simple Summary:**

Cancer has several phenotypic subtypes where the responsiveness towards drugs or capacity of migration or recurrence are different. The molecular networks are dynamically altered in various phenotypes of cancer. To reveal the network pathways in epithelial-mesenchymal transition (EMT), we have profiled gene expression in mesenchymal stem cells and diffuse-type gastric cancer (GC), as well as intestinal-type GC. Gene expression signatures revealed that the molecular pathway networks were altered in intestinal- and diffuse-type GC. The artificial intelligence (AI) recognized the differences in molecular network pictures of intestinal- and diffuse-type GC.

**Abstract:**

Epithelial-mesenchymal transition (EMT) plays an important role in the acquisition of cancer stem cell (CSC) feature and drug resistance, which are the main hallmarks of cancer malignancy. Although previous findings have shown that several signaling pathways are activated in cancer progression, the precise mechanism of signaling pathways in EMT and CSCs are not fully understood. In this study, we focused on the intestinal and diffuse-type gastric cancer (GC) and analyzed the gene expression of public RNAseq data to understand the molecular pathway regulation in different subtypes of gastric cancer. Network pathway analysis was performed by Ingenuity Pathway Analysis (IPA). A total of 2815 probe set IDs were significantly different between intestinal- and diffuse-type GC data in cBioPortal Cancer Genomics. Our analysis uncovered 10 genes including *male-specific lethal 3 homolog (Drosophila) pseudogene 1* (*MSL3P1*), *CDC28 protein kinase regulatory subunit 1B* (*CKS1B*), *DEAD-box helicase 27* (*DDX27*), *golgi to ER traffic protein 4* (*GET4*), *chromosome segregation 1 like* (*CSE1L*), *translocase of outer mitochondrial membrane 34* (*TOMM34*), *YTH N6-methyladenosine RNA binding protein 1* (*YTHDF1*), *ribonucleic acid export 1* (*RAE1*), *par-6 family cell polarity regulator beta* (*PARD6B*), and *MRG domain binding protein* (*MRGBP*), which have differences in gene expression between intestinal- and diffuse-type GC. A total of 463 direct relationships with three molecules (MYC, NTRK1, UBE2M) were found in the biomarker-filtered network generated by network pathway analysis. The networks and features in intestinal- and diffuse-type GC have been investigated and profiled in bioinformatics. Our results revealed the signaling pathway networks in intestinal- and diffuse-type GC, bringing new light for the elucidation of drug resistance mechanisms in CSCs.

## 1. Introduction

Different cell types show a variety of molecular networks. Gastric cancer (GC) has several subtypes, which includes intestinal- and diffuse-type GC [1,2]. Intestinal-type GC has a trend to be more rigid. In contrast, diffuse-type GC has a tendency to be more loose or sparse, which confers the diffuse-type GC malignant property and the migration capacity to the secondary site of cancer. It is essential to distinguish the subtypes of GC, since the prognosis is different, and the anti-cancer drug resistance may also be involved in diffuse-type GC [3]. Thus, the therapeutic strategies may differ in each subtype of GC. Although the gene mutations of *CDH1* and *RHOA* distinguished GC from colorectal and esophageal tumors, and these mutations were specific to diffuse-type GC, it is still challenging to discriminate the intestinal-type and diffuse-type GC in molecular gene expression networks [4]. We have previously revealed that the mRNA ratios of *CDH2* to *CDH1* distinguish the intestinal- and diffuse-type GC [2]. The precise molecular mechanisms behind the differences between the intestinal- and diffuse-type GC are still under investigation.

Epithelial-mesenchymal transition (EMT) is associated with the malignancy of GC and diffuse-type GC [5]. EMT is one of the critical features in cancer stem cells (CSCs), which plays an essential role in cancer metastasis and drug resistance, and therefore, is an important therapeutic target [6,7,8]. EMT program contributes to development as well as several pathogenesis conditions such as wound healing, tissue fibrosis and cancer progression [7]. Abundant molecules and networks are involved in EMT process, while core EMT transcription factors have been defined as SNAI1/2, ZEB1/2 and TWIST2 [8,9]. The EMT mechanism has many aspects and layers in morphological changes and cancer microenvironment [10]. To reveal the network pathways in the EMT, we have profiled gene expression and networks in mesenchymal stem cells and diffuse-type GC, as well as intestinal-type GC [2,11]. To better understand the pathogenesis of GC and treat EMT-like malignant diffuse-type GC, it is essential to know and predict the network pathway difference between intestinal- and diffuse-type GC.

The importance and potential to use the molecular network profile to distinguish diffuse- and intestinal-type GC are increasing in the digital era to reveal the EMT mechanism [10]. The previous study clearly demonstrated that the gene regulatory network construction identified nuclear transcription factor Y subunit alpha (NFYA) as a prognostic factor in diffuse-type GC [12]. Recent progress in computational analysis and public databases enables multi-disciplinary assessment for big data, including network analysis of the RefSeq data. In this study, the open-sourced RefSeq data of intestinal- and diffuse-type GC were compared, followed by molecular network analysis and gene ontology analysis [13]. In the meantime, the prediction modeling utilizing Artificial Intelligence (AI) for the molecular networks has been established. This research is integrating the gene expression, molecular networks and AI for future networking.

## 2. Results

### 2.1. Genes Altered in Intestinal- and Diffuse-Type GC

Genes altered in intestinal- and diffuse-type GC were analyzed in chromosomal instability (CIN) type and genomically stable (GS) type samples in TCGA RNAseq data, respectively. Table 1 shows the top 10 genes altered in intestinal- and diffuse-type GC. The top 10 genes include *male-specific lethal 3 homolog (Drosophila) pseudogene 1* (*MSL3P1*), *CDC28 protein kinase regulatory subunit 1B* (*CKS1B*), *DEAD-box helicase 27* (*DDX27*), *golgi to ER traffic protein 4* (*GET4*), *chromosome segregation 1 like* (*CSE1L*), *translocase of outer mitochondrial membrane 34* (*TOMM34*), *YTH N6-methyladenosine RNA binding protein 1* (*YTHDF1*), *ribonucleic acid export 1* (*RAE1*), *par-6 family cell polarity regulator beta* (*PARD6B*), and *MRG domain binding protein* (*MRGBP*). Gene expression profile of the top 10 genes in intestinal- and diffuse-type GC are shown in Figure 1. A total of 2815 IDs were significantly altered in intestinal- and diffuse-type GC (*t*-test, *p* < 0.00001) (Appendix A).

### 2.2. Networks Generated from Genes Altered in Intestinal- and Diffuse-Type GC

Networks of genes altered in intestinal- and diffuse-type GC were analyzed using Ingenuity Pathway Analysis (IPA). A total of 2815 IDs that had significant difference between intestinal- and diffuse-type GC were analyzed in IPA (*t*-test, *p* < 0.00001). A total of 25 networks generated from genes that have significant difference between intestinal- and diffuse-type GC are shown in Table 2. The Network #1 which is related to cancer, gastrointestinal disease, organismal injury, and abnormalities is shown in Figure 2.

### 2.3. Regulator Effect Networks Related to Cancer in Intestinal- and Diffuse-Type GC

Regulator effects were analyzed by IPA. The target disease was selected as cancer in the analysis. The types of regulators analyzed include biological drug, canonical pathway, and chemical drug (Figure 3). Table 3 shows regulator effect networks related to cancer in intestinal-type GC. Regulator effect networks related to cancer have been generated. Table 4 shows regulator effect networks related to cancer in diffuse-type GC.

### 2.4. MicroRNA (miRNA)-Related Regulator Effect Networks in Intestinal- and Diffuse-Type GC

MicroRNA (miRNA)-related regulator effect networks were analyzed in intestinal- and diffuse-type GC (Figure 4). Table 5 shows miRNA-related regulator effect networks in intestinal-type GC, whereas Table 6 shows miRNA-related regulator effect networks in diffuse-type GC.

### 2.5. Upstream Regulators in Intestinal- and Diffuse-Type GC

Upstream regulators of genes altered in intestinal- and diffuse-type GC were defined by IPA analysis. The top 25 upstream regulators of the altered genes in intestinal- and diffuse-type GC are shown in Table 7. The top 25 upstream regulators include NUPR1, CSF2, PTGER2, TP53, EGFR, let-7, ERBB2, calcitriol, RABL6, MITF, E2F1, CDKN2A, KDM1A, E2F3, EP400, BNIP3L, YAP1, MYCN, MYC, HGF, E2f, AREG, TBX2, and KDM5B.

### 2.6. Gene Ontology (GO) (Biological Process) and EMT-Related Processes of Genes Regulated in Intestinal- and Diffuse-Type GC

Gene Ontology (GO) was analyzed in genes regulated in intestinal- and diffuse-type GC. A total of 2815 IDs were analyzed for enrichment analysis in the Database for Annotations, Visualization and Integrated Discovery (DAVID) database, which resulted in 2762 DAVID gene IDs analyzed in GO Biological Process. The top 21 GOs are shown in Table 8 (modified Fischer Exact *p* value < 10^−6^, *p* < 0.005 in Bonferroni statistics). In the 2815 IDs, EMT-related genes, which have GO (Biological Process) term “epithelial to mesenchymal transition”, included *BMP and activin membrane bound inhibitor* (*BAMBI*), *EPH receptor A3* (*EPHA3*), *GLI pathogenesis related 2* (*GLIPR2*), *MAD2 mitotic arrest deficient-like 2 (yeast)* (*MAD2L2*), *SMAD family member 4* (*SMAD4*), *SRY-box 9* (*SOX9*), *Wnt family member 11* (*WNT11*), *adiponectin receptor 1* (*ADIPOR1*), *bone morphogenetic protein 7* (*BMP7*), *ephrin A1* (*EFNA1*), *forkhead box A2* (*FOXA2*), *hepatocyte growth factor* (*HGF*), *histone deacetylase 2* (*HDAC2*), *low density lipoprotein receptor class A domain containing 4* (*LDLRAD4*), *msh homeobox 2* (*MSX2*), *ovo like zinc finger 2* (*OVOL2*), *pleckstrin homology like domain family B member 1* (*PHLDB1*), *transforming growth factor beta receptor 2* (*TGFBR2*), and *tripartite motif containing 28* (*TRIM28*).

EMT may be related to the stemness of cancer. Genes related to stem cells have been investigated in the 2815 IDs, which have significant differences between intestinal- and diffuse-type GC. Genes which have the “stem cell” term in GO (Biological Process) included *CCR4-NOT transcription complex subunit 3* (*CNOT3*), *CD34 molecule* (*CD34*), *ETS variant 4* (*ETV4*), *Fanconi anemia complementation group D2* (*FANCD2*), *GATA binding protein 2* (*GATA2*), *LDL receptor related protein 5* (*LRP5*), *LIM domain binding 2* (*LDB2*), *RNA polymerase II subunit C* (*POLR2C*), *RNA polymerase II subunit H* (*POLR2H*), *RNA polymerase II subunit J* (*POLR2J*), *SMAD family member 4* (*SMAD4*), *SRY-box 2* (*SOX2*), SRY-box 9 (*SOX9*), Wnt family member 2B (*WNT2B*), *abnormal spindle microtubule assembly* (*ASPM*), *alpha-2-macroglobulin* (*A2M*), *cullin 4A* (*CUL4A*), *cyclin dependent kinase inhibitor 2A* (*CDKN2A*), *endothelial PAS domain protein 1* (*EPAS1*), *epithelial cell adhesion molecule* (*EPCAM*), *fibulin 1* (*FBLN1*), *frizzled class receptor 1* (*FZD1*), *growth arrest specific 6* (*GAS6*), *hes family bHLH transcription factor 1* (*HES1*), *insulin like growth factor 2 mRNA binding protein 1* (*IGF2BP1*), *keratinocyte differentiation factor 1* (*KDF1*), *lysine demethylase 1A* (*KDM1A*), *mediator complex subunit 10* (*MED10*), *mediator complex subunit 24* (*MED24*), *mediator complex subunit 30* (*MED30*), *mesenchyme homeobox 1* (*MEOX1*), *msh homeobox 2* (*MSX2*), *notchless homolog 1* (*NLE1*), *ovo like zinc finger 2* (*OVOL2*), *paired related homeobox 1* (*PRRX1*), *platelet activating factor acetylhydrolase 1b regulatory subunit 1* (*PAFAH1B1*), *platelet derived growth factor receptor alpha* (*PDGFRA*), *programmed cell death 2* (*PDCD2*), *proteasome 26S subunit, non-ATPase 11* (*PSMD11*), *spalt like transcription factor 4* (*SALL4*), *squamous cell carcinoma antigen recognized by T-cells 3* (*SART3*), ubiquitin associated protein 2 like (UBAP2L), and *vacuolar protein sorting 72 homolog* (*VPS72*).

Metabolism is one of the possible EMT-modified processes. Genes which have “metabolism” term in KEGG PATHWAY in the analysis of DAVID of the 2815 IDs are listed in Table 9. A total of 166 genes have been found to have “metabolism” term in KEGG PATHWAY annotation. Potential EMT-related genes with terms of CTNNB, ZEB, ERBB, TGFB, SMAD, CDH, STAT, AKT, WNT, and TWIST were searched in the 2815 IDs, which resulted in the selection of *CTNNBL1*, *ZEB2*, *ERBB3*, *TGFBR2*, *SMAD4*, *CDH5*, *STAT5A*, *AKT3*, *STAT5A*, *AKT3*, *STAT5B*, *WNT11*, *STAT1*, *WNT2B*, *ERBB2,* and *TWIST2* [10].

### 2.7. Prediction Model for Molecular Networks of Intestinal- and Diffuse-Type GC

The results of upstream analysis of intestinal- and diffuse-type GC data were analyzed in DataRobot Automated Machine Learning version 6.0 for creating prediction models. The list of upstream regulators was up-loaded and linked with network picture data, followed by the target prediction setting as subtype differences in intestinal- and diffuse-type GC (Figure 5). Among various prediction models DataRobot created, Elastic-Net Classifier (mixing alpha = 0.5/Binomial Deviance) was the highest predictive accuracy model with AUC of 0.7185 in cross-validation score. For this model, the feature impact chart using Permutation Importance showed that the most important features for accurately predicting the subtype of GC (“Analysis” values) were upstream network pictures (NWpic) (Figure 5a,b). Figure 5c shows the Partial Dependence Plot in Predicted Activation State of the upstream network. Figure 5d shows the Word Cloud of the target molecules. The size of the molecules indicates the appearance in the dataset, and the color shows the coefficient. Figure 5e shows the activation maps where the attention of AI is highlighted. Figure 5f shows an exemplified Receiver Operating Characteristic (ROC) curve for the model.

### 2.8. EMT Molecular Pathway and Diffuse-Type GC Mapping

The canonical pathways for Regulation of the EMT pathway include TGF-beta pathway, Wnt pathway, Notch pathway, and Receptor Tyrosine Kinase pathway (Figure 6). In each pathway related to EMT, genes of which expression was altered in diffuse-type GC compared to intestinal-type GC are mapped in pink (up-regulated) or green (down-regulated) color. The activation states of the pathways are predicted with IPA and shown in orange (activation) or blue (inhibition) color. RNA–RNA interaction analysis identified the interacted miRNAs as let-7, mir-10, mir-126, mir-181, mir-26, mir-515, MIR100-LET7A2-MIR125B1, MIR124, MIR99A-LET7C-MIR125B2, and MIRLET7.

## 3. Discussion

It is critical to distinguish the intestinal- and diffuse-type GC for effective therapeutic strategies, since the pathogenesis and prognosis are quite different in these subtypes. We previously revealed the gene signature of intestinal- and diffuse-type GC, which is indicated by the ratio of gene expression in *CDH2* to *CDH1* [2]. CDH1 and CDH2 are important factors as the signatures for distinguishing the subtypes of GC. Since our previous reports, the abundant useful open-source data, including RefSeq data for the intestinal- and diffuse-type GC, have been available in public [13,14,15,16]. Our current study highlights the relevance of using open-source data for human health. In the current study, the RefSeq data of intestinal- and diffuse-type GC have been analyzed for exploring the molecular networks and AI modeling application. The top 10 genes of which gene expression was altered in intestinal- and diffuse-type GC RefSeq data included *CKS1B*, *CSE1L*, *DDX27*, *GET4*, *MRGBP*, *MSL3P1*, *PARD6B*, *RAE1*, *TOMM34,* and *YTHDF1*. The network analysis of altered genes in intestinal- and diffuse-type GC generated networks related to cancer, gastrointestinal disease, organismal injury and abnormalities, amino acid metabolism, molecular transport, small molecule biochemistry, and so on. Several miRNAs including miR-205-5p, miR-21-5p, let-7a-5p, let-7, miR-24-3p, and miR-291a-3p were identified to regulate networks involved in intestinal- and diffuse-type GC. Since previous studies have revealed the involvement of miR-200s in promoting metastatic colonization by inhibiting EMT and promoting mesenchymal-epithelial transition (MET), it may be an intriguing approach to reveal miRNA networks in EMT [17,18]. The several miRNAs are involved and regulated in EMT and MET, which would be critical for progression and metastasis process [19,20,21]. DataRobot Automated Machine Learning created prediction models to distinguish intestinal- and diffuse-type GC with results of up-stream analysis and the network picture data. The image recognition of molecular networks by AI would distinguish the intestinal- and diffuse-type GC. It was indicated that Predicted Activation State could anticipate the subtypes of GC with approximately 0.5 of partial dependence, which showed that the predicted activation state of the molecular networks might distinguish the subtypes of GC.

The intestinal- and diffuse-type GC can be distinguished with the mRNA ratios of *CDH2* to *CDH1*, as previously shown [2]. The molecular network profiling is vital to reveal the mechanisms behind the differences between the intestinal- and diffuse-type GC, such as EMT and drug resistance in CSCs. The research exploring the differences between molecular networks in intestinal- and diffuse-type GC would reveal the interesting mechanisms leading to the therapeutic target identification. It is easier to detect miRNAs in the blood than to analyze the tissues. The current study exploring the miRNA regulation in intestinal- and diffuse-type GC might identify the miRNAs involving the EMT in diffuse-type GC, and these miRNAs might be detected in the blood. The profile in the molecular networks of RNAs detected in blood would be the next pathways to be revealed in future research.

## 4. Materials and Methods 

### 4.1. Data Collection

The RefSeq data of intestinal- and diffuse-type GC are publicly available in The Cancer Genome Atlas (TCGA) of The cBioPortal for Cancer Genomics database [13,14,15] in NCI Genomic Data Commons (GDC) Data Portal [22]. From the data of stomach adenocarcinoma (TCGA, PanCancer Atlas), intestinal- and diffuse-type GC data, which are noted as chromosomal instability (CIN) and genomically stable (GS), respectively, in TCGA Research Network publication, were compared [13].

### 4.2. Network Analysis

Data of intestinal- and diffuse-type GC in TCGA cBioPortal Cancer Genomics were uploaded and analyzed through the use of Ingenuity Pathway Analysis (IPA) (QIAGEN Inc., Hilden, Germany) [23].

### 4.3. Gene Ontology (GO) Analysis

Gene Ontology (GO) was analyzed in the Database for Annotations, Visualization and Integrated Discovery (DAVID) Bioinformatics Resources 6.8 (Laboratory of Human Retrovirology and Immunoinformatics) [24,25].

### 4.4. AI Prediction Modeling

To create a prediction model by using multi-modal data including images and text description of molecular networks, an enterprise AI platform (DataRobot Automated Machine Learning version 6.0; DataRobot Inc., Boston, MA, USA) was used. For the modeling, the 116 molecular networks of IPA upstream analysis in intestinal- and diffuse-type GC were collected and input as image data in the DataRobot (58 images in each subtype), which automatically created and tuned prediction models using various machine learning algorithms (e.g., eXtreme gradient-boosted trees, random forest, regularized regression such as Elastic Net, Neural Networks) [26,27]. Finally, the AI model with the highest predictive accuracy on DataRobot was identified and various insights (such as Permutation Importance or Partial Dependence Plot) obtained from the model were reviewed.

### 4.5. Data Visualization

The results of gene expression data of RefSeq and network analysis were visualized by Tableau software.

### 4.6. Statistical Analysis

The RefSeq data were analyzed by Student’s t-test. Z-score in intestinal- and diffuse-type GC samples were compared, and the difference was considered to be significant in *p* value < 0.00001. For DAVID Gene Ontology (GO) enrichment analysis, data was analyzed in the default setting. GO enrichment was considered significant in modified Fischer Exact *p* value < 10^−6^. Bonferroni statistics showed *p* value < 0.005.

## 5. Conclusions

The regulatory molecular networks are altered in intestinal- and diffuse-type GC. Networks generated from genes altered in intestinal- and diffuse-type GC included a network related to cancer, gastrointestinal disease, and organismal injury and abnormalities. We demonstrated that several miRNAs regulated the networks in intestinal- and diffuse-type GC. Machine learning of network image data created prediction models to distinguish the subtypes of the GC. The molecular mapping of intestinal- and diffuse-type GC may reveal the EMT mechanism. The miRNAs identified in the study may be regulated in EMT, which would be critical for progression and metastasis process. Our results support further identification of GC subtypes through visual changes in molecular networks.

## Figures and Tables

**Figure 1 cancers-12-03833-f001:**
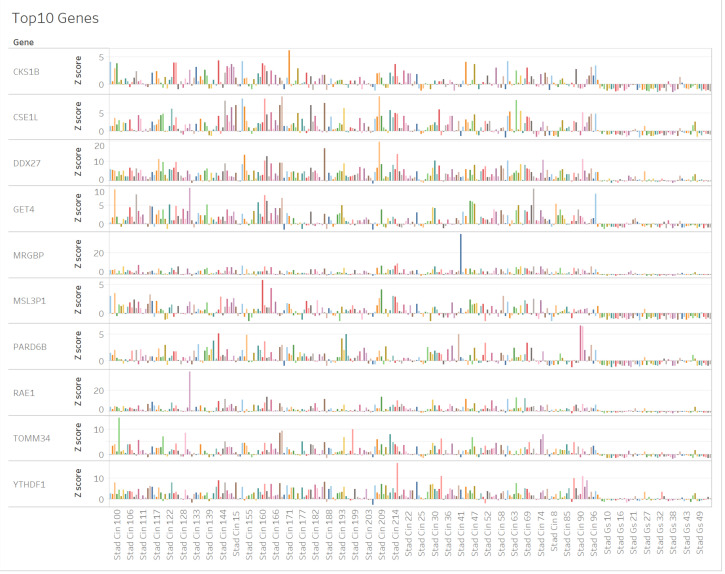
Gene expression profile of top 10 genes altered in intestinal- and diffuse-type gastric cancer (GC). The gene expression of top 10 genes, which have significant difference between intestinal-type (CIN; chromosomal instability) and diffuse-type (GS; genomically stable) gastric cancer (GC) in The Cancer Genome Atlas (TCGA); RNAseq data are shown in Tableau visualization.

**Figure 2 cancers-12-03833-f002:**
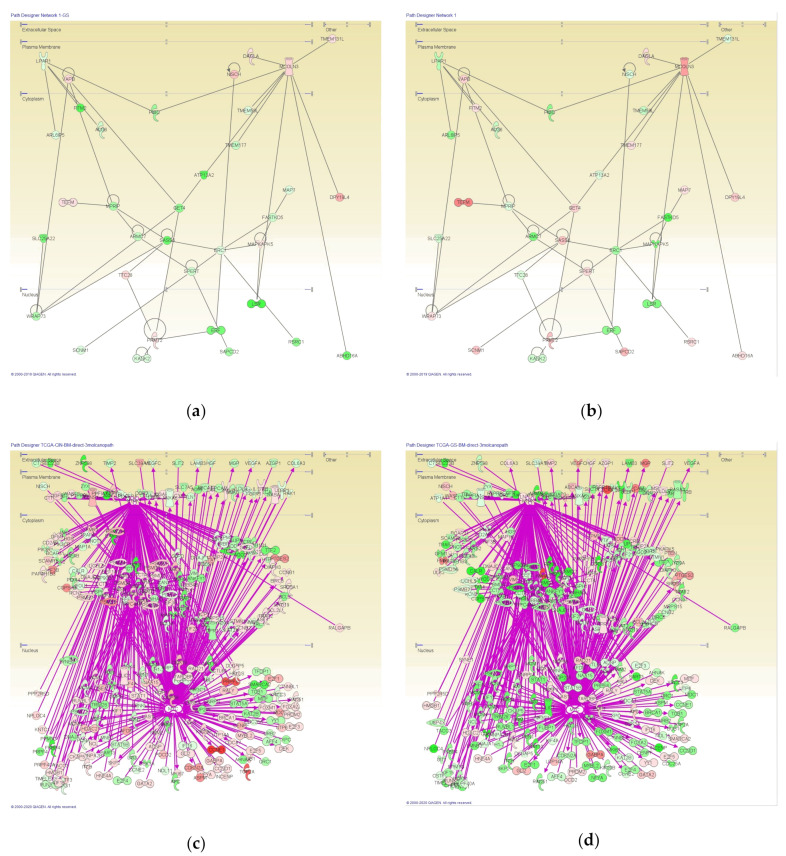
Networks generated from genes altered in intestinal- and diffuse-type gastric cancer (GC). A total of 2815 IDs, which had significant difference between intestinal- and diffuse-type GC, were analyzed in Ingenuity Pathway Analysis (IPA); and Network 1 related to cancer, gastrointestinal disease, organismal injury and abnormalities is shown. (**a**) Network in intestinal-type GC; (**b**) Network in diffuse-type GC. A total of 463 direct relationships with three molecules (MYC, NTRK1, UBE2M) are shown in the network of biomarker-filtered genes in intestinal-type GC (**c**) and diffuse-type GC (**d**). From 613 genes biomarker-filtered (human, blood, cancer), 285 genes including MYC, NTRK1 and UBE2M are included in the network. All relationships were 609.

**Figure 3 cancers-12-03833-f003:**
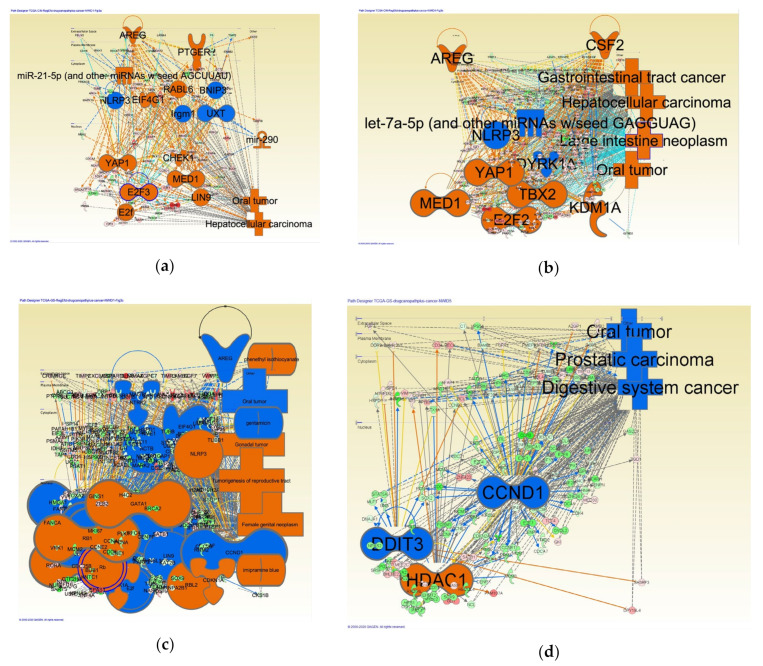
Networks for regulator effects related to cancer in intestinal- and diffuse-type gastric cancer (GC). Regulator effects were analyzed by IPA. The target disease was selected as cancer in the analysis. Type of regulators analyzed include biological drug, canonical pathway, and chemical drug. (**a**) Regulator effect network ID1 (Hepatocellular carcinoma, Oral tumor) related to cancer in intestinal-type GC; (**b**) Regulator effect network ID4 (Gastrointestinal tract cancer, Hepatocellular carcinoma, Large intestine neoplasm, Oral tumor) related to cancer in intestinal-type GC; (**c**) Regulator effect network ID1 (Female genital neoplasm, Gonadal tumor, Oral tumor, Tumorigenesis of reproductive tract) related to cancer in diffuse-type GC; (**d**) Regulator effect network ID5 (Digestive system cancer, Oral tumor, Prostatic carcinoma) related to cancer in diffuse-type GC.

**Figure 4 cancers-12-03833-f004:**
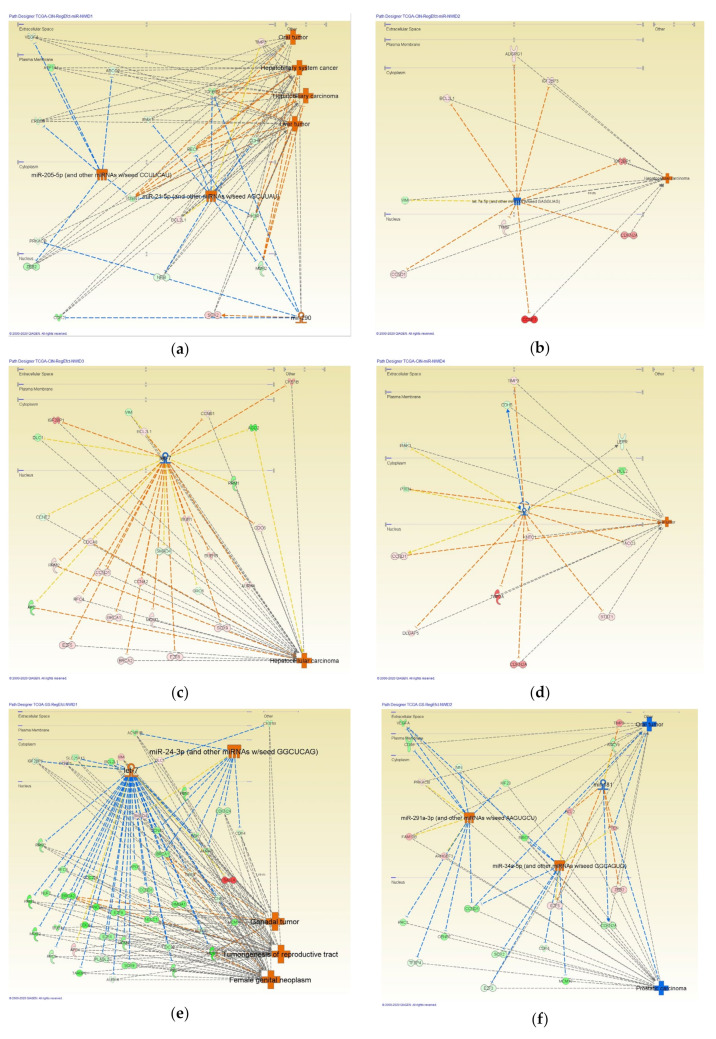
MicroRNA (miRNA)-regulated networks in intestinal- and diffuse-type gastric cancer (GC). The molecular regulators for which types were set as “miRNA” and “mature miRNA” were analyzed in the data set of intestinal-type (**a**–**d**) or diffuse-type (**e**–**i**) GC. Four networks were generated in intestinal-type GC, while five networks were generated in diffuse-type GC. (**a**) Network ID#1 regulated by miR-205-5p (and other miRNAs w/seed CCUUCAU), miR-21-5p (and other miRNAs w/seed AGCUUAU), and mir-290 in intestinal-type GC; (**b**) Network ID#2 regulated by let-7a-5p (and other miRNAs w/seed GAGGUAG) in intestinal-type GC; (**c**) Network ID#3 regulated by let-7 in intestinal-type GC; (**d**) Network ID#4 regulated by mir-21 in intestinal-type GC; (**e**) Network ID#1 regulated by let-7, miR-24-3p (and other miRNAs w/seed GGCUCAG) in diffuse-type GC; (**f**) Network ID#2 regulated by mir-181, miR-291a-3p (and other miRNAs w/seed AAGUGCU), miR-34a-5p (and other miRNAs w/seed GGCAGUG) in diffuse-type GC; (**g**) Network ID#3 regulated by mir-21 in diffuse-type GC; (**h**) Network ID#4 regulated by mir-21 in diffuse-type GC; and (**i**) Network ID#5 regulated by mir-21 in diffuse-type GC.

**Figure 5 cancers-12-03833-f005:**
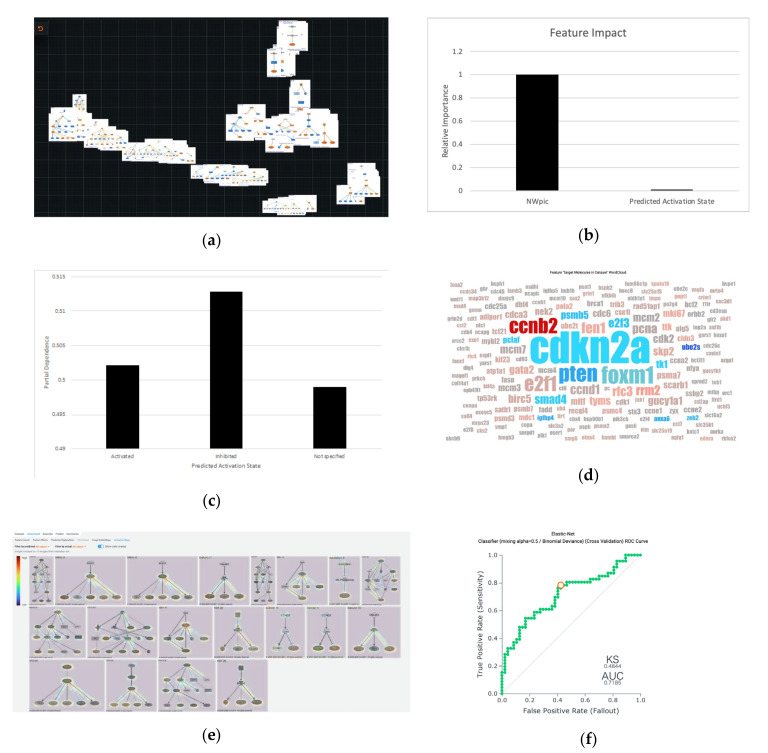
AI-oriented prediction model in intestinal- and diffuse-type gastric cancer (GC). The results of upstream analysis of intestinal- and diffuse-type GC data in IPA were analyzed in DataRobot Automated Machine Learning version 6.0 (DataRobot) for creating prediction models. The list of upstream regulators was up-loaded and linked with the network picture data, followed by the target prediction setting as subtype differences in intestinal- and diffuse-type GC. Among various prediction models DataRobot created, Elastic-Net Classifier (mixing alpha = 0.5/Binomial Deviance) was the highest predictive accuracy model with AUC of 0.7185 in cross-validation score. For this model, the feature impact chart using Permutation Importance showed that the most important features for accurately predicting the subtype of GC (“Analysis” values) were upstream network pictures (NWpic). (**a**) The Image Embedding of 93 images for creating the insight; (**b**) Feature Impact for showing the important features for predicting the subtype of GC; (**c**) The Partial Dependence Plot in Predicted Activation State; (**d**) The Word Cloud of the target molecules. The size of the molecules indicates the appearance in the dataset, and the color shows coefficient; (**e**) The activation maps where the attention of AI is highlighted; (**f**) Receiver Operating Characteristic (ROC) curve for the model.

**Figure 6 cancers-12-03833-f006:**
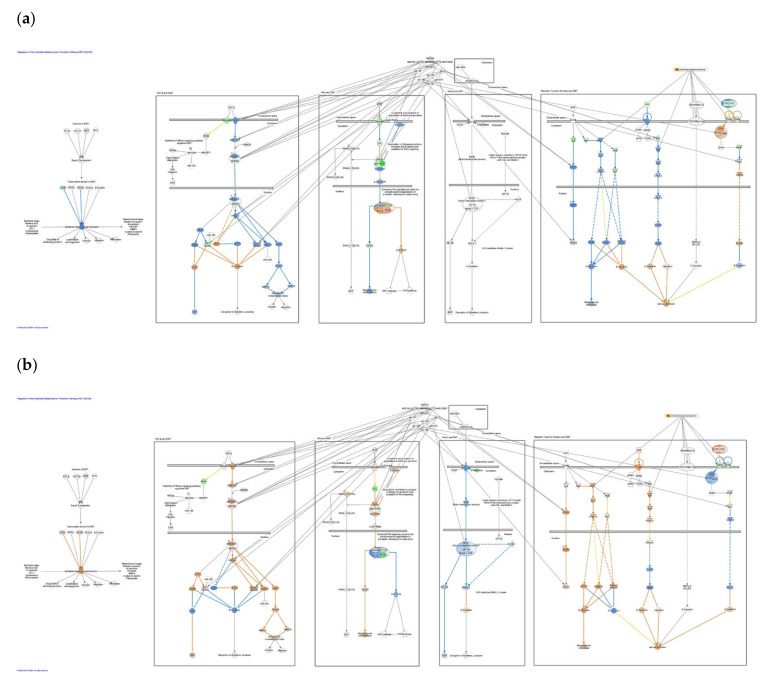
Canonical Pathways for Regulation of the EMT pathway in intestinal- and diffuse-type GC. Gene mapping and in silico prediction of the upstream and downstream effects of activation or inhibition on molecules are shown in Canonical pathways for Regulation of the EMT pathway. The genes of which expression was altered in intestinal- and diffuse-type GC are shown in pink (up-regulated) or green (down-regulated). Predicted activation or inhibition is shown in orange or blue, respectively. (**a**) Gene expression and pathway activity prediction in intestinal-type GC are shown. (**b**) Gene expression and pathway activity prediction in diffuse-type GC are shown.

**Table 1 cancers-12-03833-t001:** Top 10 genes altered in intestinal- and diffuse-type gastric cancer (GC). The top 10 genes that have significant difference between intestinal-type (CIN; chromosomal instability) and diffuse-type (GS; genomically stable) in TCGA RNAseq data are shown. A total of 2815 probe set IDs were significantly different between CIN and GS (Student’s *t*-test, *p* < 0.00001). Gene Ontology (GO) of the 10 genes are shown from the Database for Annotations, Visualization and Integrated Discovery (DAVID) analysis.

Gene Symbol	Gene Name	GOTERM_BP_DIRECT
MSL3P1	*male-specific lethal 3 homolog (Drosophila) pseudogene 1*	GO:0006338~chromatin remodeling, GO:0006342~chromatin silencing, GO:0006351~transcription, DNA-templated, GO:0016575~histone deacetylation, GO:0043967~histone H4 acetylation, GO:0043968~histone H2A acetylation,
CKS1B	*CDC28 protein kinase regulatory subunit 1B*	GO:0007049~cell cycle, GO:0007346~regulation of mitotic cell cycle, GO:0008283~cell proliferation, GO:0044772~mitotic cell cycle phase transition, GO:0045737~positive regulation of cyclin-dependent protein serine/threonine kinase activity, GO:0045893~positive regulation of transcription, DNA-templated, GO:0051301~cell division,
DDX27	*DEAD-box helicase 27*	GO:0006364~rRNA processing, GO:0010501~RNA secondary structure unwinding,
GET4	*golgi to ER traffic protein 4*	GO:0006810~transport, GO:0051220~cytoplasmic sequestering of protein, GO:0071816~tail-anchored membrane protein insertion into ER membrane, GO:1904378~maintenance of unfolded protein involved in ERAD pathway,
CSE1L	*chromosome segregation 1 like*	GO:0006606~protein import into nucleus, GO:0006611~protein export from nucleus, GO:0006915~apoptotic process, GO:0008283~cell proliferation,
TOMM34	*translocase of outer mitochondrial membrane 34*	GO:0006626~protein targeting to mitochondrion,
YTHDF1	*YTH N6-methyladenosine RNA binding protein 1*	GO:0045948~positive regulation of translational initiation,
RAE1	*ribonucleic acid export 1*	GO:0000972~transcription-dependent tethering of RNA polymerase II gene DNA at nuclear periphery, GO:0006406~mRNA export from nucleus, GO:0006409~tRNA export from nucleus, GO:0006606~protein import into nucleus, GO:0007077~mitotic nuclear envelope disassembly, GO:0010827~regulation of glucose transport, GO:0016032~viral process, GO:0016925~protein sumoylation, GO:0019083~viral transcription, GO:0031047~gene silencing by RNA, GO:0071407~cellular response to organic cyclic compound, GO:0075733~intracellular transport of virus, GO:1900034~regulation of cellular response to heat,
PARD6B	*par-6 family cell polarity regulator beta*	GO:0006461~protein complex assembly, GO:0007043~cell-cell junction assembly, GO:0007049~cell cycle, GO:0007163~establishment or maintenance of cell polarity, GO:0007409~axonogenesis, GO:0030334~regulation of cell migration, GO:0051301~cell division, GO:0070830~bicellular tight junction assembly,
MRGBP	*MRG domain binding protein*	GO:0006351~transcription, DNA-templated, GO:0006357~regulation of transcription from RNA polymerase II promoter, GO:0016573~histone acetylation, GO:0040008~regulation of growth,

**Table 2 cancers-12-03833-t002:** Networks generated from genes which have significant difference between intestinal- and diffuse-type gastric cancer (GC). The networks were generated from a total of 2815 probe set IDs differentiated between intestinal-type (CIN; chromosomal instability) and diffuse-type (GS; genomically stable) GC (Student’s t-test, *p* < 0.00001).

ID	Focus Molecules	Top Diseases and Functions
1	35	Cancer, Gastrointestinal Disease, Organismal Injury and Abnormalities
2	35	Amino Acid Metabolism, Molecular Transport, Small Molecule Biochemistry
3	34	Cardiovascular Disease, Gene Expression, Protein Synthesis
4	34	Developmental Disorder, Hereditary Disorder, Neurological Disease
5	34	Dental Disease, Dermatological Diseases and Conditions, Post-Translational Modification
6	34	Hereditary Disorder, Infectious Diseases, RNA Post-Transcriptional Modification
7	34	Carbohydrate Metabolism, Lipid Metabolism, Post-Translational Modification
8	34	Connective Tissue Disorders, Developmental Disorder, Hereditary Disorder
9	34	Cell Cycle, Molecular Transport, Protein Trafficking
10	33	Connective Tissue Disorders, Dermatological Diseases and Conditions, Developmental Disorder
11	33	Cell Morphology, Cellular Assembly and Organization, Cellular Function and Maintenance
12	33	Gene Expression, Post-Translational Modification, RNA Damage and Repair
13	33	Cell Cycle, Cellular Growth and Proliferation, Reproductive System Development and Function
14	32	Infectious Diseases, Molecular Transport, Post-Translational Modification
15	32	Cell Cycle, Cellular Assembly and Organization, DNA Replication, Recombination, and Repair
16	32	Developmental Disorder, Hereditary Disorder, Molecular Transport
17	32	Carbohydrate Metabolism, Nucleic Acid Metabolism, Small Molecule Biochemistry
18	31	Cellular Assembly and Organization, Cellular Response to Therapeutics, DNA Replication, Recombination, and Repair
19	31	Developmental Disorder, Lipid Metabolism, Small Molecule Biochemistry
20	31	Cell Morphology, Cellular Assembly and Organization, Skeletal and Muscular System Development and Function
21	31	Cancer, Cellular Assembly and Organization, Skeletal and Muscular Disorders
22	31	Cell Cycle, Cellular Assembly and Organization, Cellular Compromise
23	31	Molecular Transport, RNA Post-Transcriptional Modification, RNA Trafficking
24	31	Nervous System Development and Function, Neurological Disease, Organ Morphology
25	31	Gene Expression, Neurological Disease, Organismal Functions

**Table 3 cancers-12-03833-t003:** Regulator effect networks related to cancer in intestinal-type gastric cancer (GC). Regulator effect networks related to cancer have been generated. Type of regulators include biological drug, canonical pathway, and chemical drug.

ID	Regulators	Target Total	Diseases & Functions
1	AREG, BNIP3L, CHEK1, E2f, E2F3, EIF4G1, Irgm1, LIN9, MED1, miR-21-5p (and other miRNAs w/seed AGCUUAU), mir-290, NLRP3, PTGER2, RABL6, UXT, YAP1	94	Hepatocellular carcinoma, Oral tumor
2	AREG, ERG, KDM5B, MIR17HG, TFDP1, YAP1	123	Hepatocellular carcinoma, Intestinal cancer, Large intestine neoplasm
3	AREG, KDM5B, miR-21-5p (and other miRNAs w/seed AGCUUAU), mir-290, MIR17HG, PTGER2, SMARCB1, TCF3, UXT, YAP1	70	Hepatocellular carcinoma
4	AREG, CSF2, DYRK1A, E2F2, KDM1A, let-7a-5p (and other miRNAs w/seed GAGGUAG), MED1, NLRP3, TBX2, YAP1	200	Gastrointestinal tract cancer, Hepatocellular carcinoma, Large intestine neoplasm, Oral tumor
5	MYCN	3	Cell death of osteosarcoma cells
6	EGFR, ERBB2, HRAS, miR-205-5p (and other miRNAs w/seed CCUUCAU), tanespimycin, tazemetostat, YAP1	57	Oral tumor
7	calcitriol, medroxyprogesterone acetate	112	Gastrointestinal adenocarcinoma, Intestinal carcinoma
8	TP53	298	Gastrointestinal carcinoma
9	5-fluorouracil	28	Liver tumor
10	TAL1	31	Liver tumor
11	NUPR1	25	Hepatocellular carcinoma
12	MITF	20	Hepatocellular carcinoma
13	26s Proteasome	23	Liver tumor
14	EP400	19	Liver tumor
15	CDKN2A	69	Intestinal cancer, Large intestine neoplasm
16	FOXO1	45	Hepatobiliary system cancer
17	E2F1	47	Hepatocellular carcinoma
18	HGF	35	Hepatocellular carcinoma
19	arsenic trioxide	32	Liver tumor
20	let-7	27	Hepatocellular carcinoma
21	TP73	36	Hepatobiliary system cancer
22	mir-21	13	Oral tumor
23	valproic acid	12	Cell death of osteosarcoma cells

**Table 4 cancers-12-03833-t004:** Regulator effect networks related to cancer in diffuse-type gastric cancer (GC). Regulator effect networks related to cancer have been generated. Type of regulators include biological drug, canonical pathway, and chemical drug.

ID	Regulators	Target Total	Diseases & Functions
1	ACTB, AREG, BRD4, CCND1, CDKN1A, DYRK1A, E2f, E2F3, EIF4G1, EWSR1, FOXM1, GATA1, gentamicin, imipramine blue, LIN9, MED1, MYCN, NLRP3, NTRK2, phenethyl isothiocyanate, Rb, RB1, RBL2, TCF3, TFDP1	276	Female genital neoplasm, Gonadal tumor, Oral tumor, Tumorigenesis of reproductive tract
2	ATF4, ATF6, BNIP3L, E2f, EIF4G1, epothilone B, ERG, FOXM1, GATA1, gentamicin, imipramine blue, Irgm1, KDM5B, let-7, miR-24-3p (and other miRNAs w/seed GGCUCAG), NLRP3, phenethyl isothiocyanate, RABL6, Rb, RB1, RBL1, RBL2, SMARCB1, ZNF281	231	Cell death of osteosarcoma cells, Female genital neoplasm, Gonadal tumor, Tumorigenesis of reproductive tract
3	alvespimycin, decitabine, EGFR, EWSR1, gentamicin, KAT6A, miR-34a-5p (and other miRNAs w/seed GGCAGUG), phenethyl isothiocyanate, SYVN1, tazemetostat, YAP1	67	Oral tumor
4	alvespimycin, calcitriol, decitabine, E2F2, EGFR, ERBB2, estrogen, EWSR1, mir-181, phenethyl isothiocyanate, tazemetostat, Vegf, YAP1	210	Oral tumor, Prostatic carcinoma
5	CCND1, DDIT3, HDAC1	140	Digestive system cancer, Oral tumor, Prostatic carcinoma
6	ATF4, ATF6, EIF4G1, EP400, FOXM1, gentamicin, Irgm1, MYC, NELFA, NELFCD, NELFE, NLRP3, ZNF281	94	Ovarian tumor
7	KDM4C, UXT	18	Frequency of tumor, Incidence of tumor
8	CSF2, DDIT3, ERG, ESR1, miR-291a-3p (and other miRNAs w/seed AAGUGCU)	131	Prostatic carcinoma
9	TP53	131	Prostatic carcinoma
10	E2F1	87	Tumorigenesis of reproductive tract
11	HGF	67	Female genital neoplasm
12	TBX2	38	Digestive system cancer
13	SMARCB1	39	Abdominal carcinoma
14	TP63	42	Tumorigenesis of reproductive tract
15	PTGER2	40	Abdominal carcinoma
16	MITF	33	Female genital neoplasm
17	mir-21	27	Prostatic carcinoma
18	NFE2L2	23	Gonadal tumor
19	CD3	19	Oral tumor
20	DNMT3B	18	Female genital neoplasm
21	cephaloridine	18	Tumorigenesis of reproductive tract
22	CDKN2A	47	Tumorigenesis of reproductive tract
23	NFE2L1	6	Cell death of osteosarcoma cells
24	EIF4E	11	Ovarian tumor
25	5-fluorouracil	5	Cell death of osteosarcoma cells
26	mibolerone	9	Ovarian tumor
27	KDM1A	25	Female genital neoplasm, Tumorigenesis of reproductive tract
28	TRAP1	6	Oral tumor
29	fulvestrant	45	Female genital neoplasm
30	NCOA3	15	Tumorigenesis of reproductive tract
31	MEF2D	9	Female genital neoplasm, Tumorigenesis of reproductive tract

**Table 5 cancers-12-03833-t005:** MicroRNA (miRNA)-related regulator effect networks in intestinal-type gastric cancer (GC).

ID	Regulators	Target Molecules in Dataset	Diseases & Functions	Known Regulator-Disease/Function Relationship
1	miR-205-5p (and other miRNAs w/seed CCUUCAU), miR-21-5p (and other miRNAs w/seed AGCUUAU), mir-290	ABCC2, ATP1A1, BCL2L1, CDH5, CDK2, ERBB3, IRAK1, MSH2, NFIB, PIK3R1, PRKACB, PTEN, RECK, SOX2, TGFBR2, TIMP3, VEGFA, ZEB2	Hepatobiliary carcinoma, Hepatobiliary system cancer, Liver tumor, Oral tumor	42% (5/12)
2	let-7a-5p (and other miRNAs w/seed GAGGUAG)	ADGRG1, BCL2L1, CCND1, CCNE1, CDKN2A, IGF2BP1, IGF2BP3, TYMS, VIM	Hepatocellular carcinoma	100% (1/1)
3	let-7	AGO2, APC, AURKA, BCL2L1, BRCA1, BRCA2, BUB1, BUB1B, CCNA2, CCNB1, CCND1, CCNE2, CDC6, CDCA8, CKS1B, DLC1, E2F5, E2F8, IGF2BP1, MCM2, ORC6, RFC4, RRM1, RRM2, SMAD4, SOX9, VIM	Hepatocellular carcinoma	100% (1/1)
4	mir-21	BCL2, CCND1, CDH5, CDKN2A, DLGAP5, IRAK1, KNTC1, LEPR, PTEN, STAT1, TACC3, TIMP3, TOP2A	Oral tumor	0% (0/1)

**Table 6 cancers-12-03833-t006:** MicroRNA (miRNA)-related regulator effect networks in diffuse-type gastric cancer (GC).

ID	Regulators	Target Molecules in Dataset	Diseases & Functions	Known Regulator-Disease/Function Relationship
1	let-7, miR-24-3p (and other miRNAs w/seed GGCUCAG)	ACVR1B, APC, AURKA, AURKB, BCL2L1, BRCA1, BRCA2, BUB1, BUB1B, CCNA2, CCNB1, CCND1, CCNE2, CDC20, CDC25A, CDC6, CDK1, CDK4, CDKN2A, CKS1B, DBF4, DLC1, E2F4, E2F8, FANCD2, FBL, FEN1, HMGA1, IGF2BP1, MCM10, MCM2, MCM7, MCM8, NOLC1, NUF2, PLAGL2, RFC4, RFC5, RRM1, RRM2, SALL4, SLC25A13, SMAD4, SOX9, TARBP2, VIM, XPO5	Female genital neoplasm, Gonadal tumor, Tumorigenesis of reproductive tract	50% (3/6)
2	mir-181, miR-291a-3p (and other miRNAs w/seed AAGUGCU), miR-34a-5p (and other miRNAs w/seed GGCAGUG)	ADCY9, ARHGEF3, BCL2, BIRC5, CCND1, CD46, CDK4, CDKN2A, CENPF, E2F3, E2F5, FAM13B, KIF23, MCM10, NIN, PRC1, PRKACB, PTEN, SOX2, TFAP4, TIMP3, VEGFA, ZEB2	Oral tumor, Prostatic carcinoma	0% (0/6)
3	mir-21	ANLN, ARL6IP1, ASPM, ATAD2, BCL2, CCNB1, CCND1, CDH5, CDKN2A, CKAP5, CSE1L, KIF23, KNTC1, LEPR, MKI67, NCAPD2, NUSAP1, PIP4K2A, PRC1, PTEN, SOX2, STAT1, TAP1, TBC1D1, TOP2A, YY1, ZWILCH	Prostatic carcinoma	0% (0/1)
4	mir-21	ANLN, ARL6IP1, ASPM, ATAD2, BCL2, C1R, CACYBP, CANX, CCNA2, CCNB1, CCND1, CDC25A, CDH5, CDKN2A, CKAP5, CKS2, CLPB, CSE1L, ECT2, FUBP1, GTSE1, HNRNPA2B1, IFI16, IRAK1, KIF23, KIF4A, KIFC1, KNTC1, LEPR, MKI67, MSH2, NCAPD2, NME1, NPAS2, NUSAP1, PIP4K2A, PRC1, PTEN, RACGAP1, RAD51AP1, RECK, SMC2, SOX2, STAT1, STMN1, TACC3, TAP1, TBC1D1, TCF21, TIMP3, TLR1, TMEM97, TOP2A, TP53RK, UBA7, VRK1, YWHAB, YY1, ZW10, ZWILCH	Frequency of tumor	100% (1/1)
5	mir-21	ANLN, ARL6IP1, ASPM, ATAD2, BCL2, C1R, CACYBP, CANX, CCNA2, CCNB1, CCND1, CDC25A, CDH5, CDKN2A, CKAP5, CKS2, CLPB, CSE1L, ECT2, FUBP1, GTSE1, HNRNPA2B1, IFI16, IRAK1, KIF23, KIF4A, KIFC1, KNTC1, LEPR, MKI67, MSH2, NCAPD2, NME1, NPAS2, NUSAP1, PIP4K2A, PRC1, PTEN, RACGAP1, RAD51AP1, RECK, SMC2, SOX2, STAT1, STMN1, TACC3, TAP1, TBC1D1, TCF21, TIMP3, TLR1, TMEM97, TOP2A, TP53RK, UBA7, VRK1, YWHAB, YY1, ZW10, ZWILCH	Incidence of tumor	100% (1/1)

**Table 7 cancers-12-03833-t007:** Upstream regulators in intestinal- and diffuse-type gastric cancer (GC) (Top 25 regulators).

Upstream Regulators	TCGA CIN	TCGA GS
NUPR1	−4.457	6.685
CSF2	4.849	−6.057
PTGER2	4.427	−5.06
TP53	−4.044	5.394
EGFR	3.75	−5.207
let-7	−3.031	5.836
ERBB2	2.986	−5.804
calcitriol	−3.349	5.194
RABL6	3.28	−5.154
MITF	2.927	−5.436
E2F1	2.141	−5.933
CDKN2A	−2.944	5
KDM1A	3.328	−4.551
E2F3	2.496	−5.334
EP400	3.183	−4.482
BNIP3L	−3.714	3.571
YAP1	3.103	−4.161
MYCN	4.044	−2.997
MYC	1.087	−5.862
HGF	2.874	−4.014
E2f	2.984	−3.881
AREG	3.525	−3.213
TBX2	2.619	−4.104
KDM5B	−4.075	2.537

**Table 8 cancers-12-03833-t008:** Gene Ontology (GO) (Biological Process) of genes regulated in intestinal- and diffuse-type gastric cancer (GC). The total 2815 probe set IDs were analyzed for enrichment analysis in DAVID, which resulted in 2394 genes analyzed in GO Biological Process. Category of GOTERM_BP_DIRECT is listed.

Term	Count
GO:0051301~cell division	121
GO:0007062~sister chromatid cohesion	54
GO:0007067~mitotic nuclear division	91
GO:0006260~DNA replication	67
GO:0031145~anaphase-promoting complex-dependent catabolic process	40
GO:0051436~negative regulation of ubiquitin-protein ligase activity involved in mitotic cell cycle	37
GO:0000082~G1/S transition of mitotic cell cycle	44
GO:0051437~positive regulation of ubiquitin-protein ligase activity involved in regulation of mitotic cell cycle transition	36
GO:0006281~DNA repair	74
GO:0006521~regulation of cellular amino acid metabolic process	27
GO:0006270~DNA replication initiation	20
GO:0043488~regulation of mRNA stability	39
GO:0006364~rRNA processing	62
GO:0007059~chromosome segregation	29
GO:0031047~gene silencing by RNA	39
GO:0038061~NIK/NF-kappaB signaling	27
GO:0060071~Wnt signaling pathway, planar cell polarity pathway	33
GO:0002479~antigen processing and presentation of exogenous peptide antigen via MHC class I, TAP-dependent	26
GO:0007077~mitotic nuclear envelope disassembly	21
GO:0000398~mRNA splicing, via spliceosome	60
GO:0070125~mitochondrial translational elongation	31

**Table 9 cancers-12-03833-t009:** Genes that have “metabolism” term in KEGG PATHWAY annotation in DAVID analysis. The total 166 genes were selected as metabolism-related genes in KEGG PATHWAY annotation as the result of DAVID analysis of the total 2815 probe set IDs.

Genes
*AGPAT2*	*BCAT2*	*HPRT1*	*PAFAH1B1*
*HACD3*	*CERS2*	*IMPDH1*	*PDGFRA*
*HACD4*	*CHDH*	*ITPA*	*PNPT1*
*BDH2*	*CHKA*	*ITPKA*	*PRIM1*
*ATIC*	*CDO1*	*ITPKB*	*PRIM2*
*ADPRM*	*CYP1B1*	*IDH1*	*PTGES2*
*AKT3*	*CYP2U1*	*IDH3B*	*PTGES3*
*CTPS1*	*DGUOK*	*LTC4S*	*PTGS1*
*CTPS2*	*DTYMK*	*LPIN3*	*PRKCB*
*DNMT3B*	*DGAT2*	*LIPT2*	*PRUNE1*
*POLA2*	*DMGDH*	*MDH2*	*PNPO*
*POLD2*	*ENTPD1*	*MARS2*	*PYCR1*
*POLE2*	*ENTPD6*	*MARS*	*PYCR2*
*POLE3*	*ENOPH1*	*MMAB*	*PC*
*HDDC3*	*ERBB2*	*MAPK10*	*RAC3*
*NANP*	*FAM213B*	*NPR1*	*RRM1*
*NFS1*	*FASN*	*NPR2*	*RRM2*
*NME1*	*FGFR1*	*NEU1*	*RPIA*
*POLR1B*	*FGFR3*	*NIT2*	*RPE*
*POLR1C*	*FLAD1*	*NUDT5*	*SEPHS1*
*POLR2C*	*FTCD*	*PTEN*	*SELENOI*
*POLR2H*	*FAH*	*PCYT2*	*SRR*
*POLR2J*	*GAL3ST1*	*PEMT*	*SLC1A5*
*POLR3C*	*GGCT*	*PIK3CB*	*SLC44A5*
*POLR3E*	*GGT5*	*PIP5K1A*	*SLC7A5*
*POLR3F*	*GGT7*	*PIP5K1C*	*SORD*
*POLR3GL*	*GNPDA1*	*PIP4K2A*	*SRM*
*POLR3K*	*G6PC3*	*PIP4K2C*	*SMOX*
*TWISTNB*	*G6PD*	*PTDSS1*	*SMPD4*
*UGT8*	*GANC*	*PDE11A*	*TAZ*
*UXS1*	*GBA*	*PDE1A*	*TXNDC12*
*WASF2*	*GUSB*	*PDE2A*	*TXNRD1*
*ACACB*	*GAD1*	*PDE4B*	*TXNRD2*
*ACSS1*	*GCLM*	*PDE5A*	*TK1*
*ACYP1*	*EPRS*	*PDE7B*	*TYMS*
*AHCY*	*EARS2*	*PGP*	*TPO*
*ADCY4*	*GSTM5*	*PIK3R1*	*TALDO1*
*ADCY9*	*GSTO2*	*PLPP1*	*UMPS*
*AK2*	*GSS*	*PPAT*	*UCKL1*
*ADH1B*	*GCSH*	*PSAT1*	*ZNRD1*
*ALDH1A1*	*GLO1*	*PSPH*	
*ASPA*	*GMPS*	*PAFAH1B3*

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
