# Peer review of "Molecular Network Profiling in Intestinal- and Diffuse-Type Gastric Cancer"

_cancers, 2020, doi:10.3390/cancers12123833_

Round 1
Reviewer 1 Report
The rationale behind this paper remains unclear. The topic is, as stated by authors, the EMT process. They performed a bioinformatics analysis of TCGA data in GC but it is unclear if there are some differences in the expression of EMT markers between the two different subtypes. More emphasis should be placed on this aspect. I suggest selecting a list of differentially expressed EMT markers using classical EMT markers (i.e. CDH1, Zeb1, Zeb2, Snail, etc...) and possible EMT modified processes (i.e. metabolism, stemness, invasion). Some articles on this topic should be also considered and cited: - Front Med. 2018 Aug;12(4):361-373. / Nat Rev Mol Cell Biol. 2020 Jun;21(6):341-352. / Semin Cancer Biol. 2019 Oct;58:1-10. / Front Oncol. 2017 Nov 13;7:274. In my opinion, major revisions are still necessary.
Author Response
Thank you for your constructive suggestions. Following your comment, genes related to classical EMT markers and possible EMT modified process has been analyzed on a list of differentially expressed EMT markers. The classical EMT markers such as CDH1 and ZEB1 were not included in the 2815 IDs, which were significantly different in Z-score of intestinal- and diffuse-type GC, although ZEB2 was included. The p-value in CDH1 is p=0.00002210, which was above the current threshold of p<0.00001, seems to have significant difference. Instead, the potential EMT-related genes with terms of CTNNB, ZEB, ERBB, TGFB, SMAD, CDH, STAT, AKT, WNT and TWIST were searched in the 2815 IDs. Descriptions related to EMT and stemness have been added in the section 2.6 Gene Ontology (GO) (Biological Process) and EMT-related processes of genes regulated in intestinal- and diffuse-type GC, and Table 9 as “Genes which have “metabolism” term in KEGG PATHWAY annotation in DAVID analysis” has been added. Search with “invasion” has not resulted in the reasonable extraction and processes with terms of “bacterial invasion” were found:
<line 220-255>
In the 2815 IDs, EMT-related genes which have GO (Biological Process) term “epithelial to mesenchymal transition” included BMP and activin membrane bound inhibitor (BAMBI), EPH receptor A3 (EPHA3), GLI pathogenesis related 2 (GLIPR2), MAD2 mitotic arrest deficient-like 2 (yeast) (MAD2L2), SMAD family member 4 (SMAD4), SRY-box 9 (SOX9), Wnt family member 11 (WNT11), adiponectin receptor 1(ADIPOR1), bone morphogenetic protein 7 (BMP7), ephrin A1 (EFNA1), forkhead box A2 (FOXA2), hepatocyte growth factor (HGF), histone deacetylase 2 (HDAC2), low density lipoprotein receptor class A domain containing 4 (LDLRAD4), msh homeobox 2 (MSX2), ovo like zinc finger 2(OVOL2), pleckstrin homology like domain family B member 1 (PHLDB1), transforming growth factor beta receptor 2 (TGFBR2), tripartite motif containing 28 (TRIM28).
EMT may be related to the stemness of cancer. Genes related to stem cells have been investigated in the 2815 IDs which have significant differences between intestinal- and diffuse-type GC. Genes which have “stem cell” term in GO (Biological Process) included CCR4-NOT transcription complex subunit 3 (CNOT3), CD34 molecule (CD34), ETS variant 4 (ETV4), Fanconi anemia complementation group D2(FANCD2), GATA binding protein 2 (GATA2), LDL receptor related protein 5 (LRP5), LIM domain binding 2 (LDB2), RNA polymerase II subunit C (POLR2C), RNA polymerase II subunit H (POLR2H), RNA polymerase II subunit J (POLR2J), SMAD family member 4 (SMAD4), SRY-box 2(SOX2), SRY-box 9 (SOX9), Wnt family member 2B (WNT2B), abnormal spindle microtubule assembly (ASPM), alpha-2-macroglobulin (A2M),cullin 4A (CUL4A), cyclin dependent kinase inhibitor 2A (CDKN2A), endothelial PAS domain protein 1 (EPAS1), epithelial cell adhesion molecule(EPCAM), fibulin 1 (FBLN1), frizzled class receptor 1 (FZD1), growth arrest specific 6 (GAS6), hes family bHLH transcription factor 1 (HES1), insulin like growth factor 2 mRNA binding protein 1 (IGF2BP1), keratinocyte differentiation factor 1 (KDF1), lysine demethylase 1A (KDM1A),mediator complex subunit 10 (MED10), mediator complex subunit 24 (MED24), mediator complex subunit 30 (MED30), mesenchyme homeobox 1(MEOX1), msh homeobox 2 (MSX2), notchless homolog 1 (NLE1), ovo like zinc finger 2 (OVOL2), paired related homeobox 1 (PRRX1), platelet activating factor acetylhydrolase 1b regulatory subunit 1 (PAFAH1B1), platelet derived growth factor receptor alpha (PDGFRA), programmed cell death 2 (PDCD2), proteasome 26S subunit, non-ATPase 11 (PSMD11), spalt like transcription factor 4 (SALL4), squamous cell carcinoma antigen recognized by T-cells 3 (SART3), ubiquitin associated protein 2 like (UBAP2L), vacuolar protein sorting 72 homolog (VPS72).
Metabolism is one of the possible EMT-modified processes. Genes which have “metabolism” term in KEGG PATHWAY in the analysis of DAVID of the 2815 IDs are listed in Table 9. Total 166 genes have been found to have “metabolism” term in KEGG PATHWAY annotation. Potential EMT-related genes with terms of CTNNB, ZEB, ERBB, TGFB, SMAD, CDH, STAT, AKT, WNT and TWIST were searched in the 2815 IDs, which resulted in the selection of CTNNBL1, ZEB2, ERBB3, TGFBR2, SMAD4, CDH5, STAT5A, AKT3, STAT5A, AKT3, STAT5B, WNT11, STAT1, WNT2B, ERBB2 and TWIST2 [10].
<line 261-264>
Table 9. Genes which have “metabolism” term in KEGG PATHWAY annotation in DAVID analysis. The total 166 genes were selected as metabolism-related genes in KEGG PATHWAY annotation as the result of DAVID analysis of total 2815 probe set IDs.
|
Genes |
||||
|
AGPAT2 |
BCAT2 |
HPRT1 |
PAFAH1B1 |
|
|
HACD3 |
CERS2 |
IMPDH1 |
PDGFRA |
|
|
HACD4 |
CHDH |
ITPA |
PNPT1 |
|
|
BDH2 |
CHKA |
ITPKA |
PRIM1 |
|
|
ATIC |
CDO1 |
ITPKB |
PRIM2 |
|
|
ADPRM |
CYP1B1 |
IDH1 |
PTGES2 |
|
|
AKT3 |
CYP2U1 |
IDH3B |
PTGES3 |
|
|
CTPS1 |
DGUOK |
LTC4S |
PTGS1 |
|
|
CTPS2 |
DTYMK |
LPIN3 |
PRKCB |
|
|
DNMT3B |
DGAT2 |
LIPT2 |
PRUNE1 |
|
|
POLA2 |
DMGDH |
MDH2 |
PNPO |
|
|
POLD2 |
ENTPD1 |
MARS2 |
PYCR1 |
|
|
POLE2 |
ENTPD6 |
MARS |
PYCR2 |
|
|
POLE3 |
ENOPH1 |
MMAB |
PC |
|
|
HDDC3 |
ERBB2 |
MAPK10 |
RAC3 |
|
|
NANP |
FAM213B |
NPR1 |
RRM1 |
|
|
NFS1 |
FASN |
NPR2 |
RRM2 |
|
|
NME1 |
FGFR1 |
NEU1 |
RPIA |
|
|
POLR1B |
FGFR3 |
NIT2 |
RPE |
|
|
POLR1C |
FLAD1 |
NUDT5 |
SEPHS1 |
|
|
POLR2C |
FTCD |
PTEN |
SELENOI |
|
|
POLR2H |
FAH |
PCYT2 |
SRR |
|
|
POLR2J |
GAL3ST1 |
PEMT |
SLC1A5 |
|
|
POLR3C |
GGCT |
PIK3CB |
SLC44A5 |
|
|
POLR3E |
GGT5 |
PIP5K1A |
SLC7A5 |
|
|
POLR3F |
GGT7 |
PIP5K1C |
SORD |
|
|
POLR3GL |
GNPDA1 |
PIP4K2A |
SRM |
|
|
POLR3K |
G6PC3 |
PIP4K2C |
SMOX |
|
|
TWISTNB |
G6PD |
PTDSS1 |
SMPD4 |
|
|
UGT8 |
GANC |
PDE11A |
TAZ |
|
|
UXS1 |
GBA |
PDE1A |
TXNDC12 |
|
|
WASF2 |
GUSB |
PDE2A |
TXNRD1 |
|
|
ACACB |
GAD1 |
PDE4B |
TXNRD2 |
|
|
ACSS1 |
GCLM |
PDE5A |
TK1 |
|
|
ACYP1 |
EPRS |
PDE7B |
TYMS |
|
|
AHCY |
EARS2 |
PGP |
TPO |
|
|
ADCY4 |
GSTM5 |
PIK3R1 |
TALDO1 |
|
|
ADCY9 |
GSTO2 |
PLPP1 |
UMPS |
|
|
AK2 |
GSS |
PPAT |
UCKL1 |
|
|
ADH1B |
GCSH |
PSAT1 |
ZNRD1 |
|
|
ALDH1A1 |
GLO1 |
PSPH |
|
|
|
ASPA |
GMPS |
PAFAH1B3 |
|
|
Suggested references have been cited and descriptions have been added in Introduction as follows:
<line 66-75>
The precise molecular mechanisms behind the differences between the intestinal- and diffuse-type GC have still been in an investigation.
Epithelial-mesenchymal transition (EMT) is associated with the malignancy of GC and diffuse-type GC [5]. EMT is one of the critical features in cancer stem cells (CSCs), which plays an essential role in cancer metastasis and drug resistance, therefore is an important therapeutic target [6-8]. EMT program contributes to development, and several pathogenesis conditions such as wound healing, tissue fibrosis and cancer progression as well [7]. Abundant molecules and networks are involved in EMT process, while core EMT transcription factors have been defined as SNAI1/2, ZEB1/2 and TWIST2 [8, 9]. The EMT mechanism has many aspects and layers in morphological changes and cancer microenvironment [10].
<line 80-81>
The importance and potential to use the molecular network profile to distinguish diffuse- and intestinal-type GC are increasing in the digital era to reveal the EMT mechanism [10].
<line 446-455>
- Zhang, Y.; Weinberg, R.A. Epithelial-to-mesenchymal transition in cancer: complexity and opportunities. Front Med 2018, 12, 361-373, doi:10.1007/s11684-018-0656-6.
- Yang, J.; Antin, P.; Berx, G.; Blanpain, C.; Brabletz, T.; Bronner, M.; Campbell, K.; Cano, A.; Casanova, J.; Christofori, G., et al. Guidelines and definitions for research on epithelial-mesenchymal transition. Nat Rev Mol Cell Biol 2020, 21, 341-352, doi:10.1038/s41580-020-0237-9.
- Simeone, P.; Trerotola, M.; Franck, J.; Cardon, T.; Marchisio, M.; Fournier, I.; Salzet, M.; Maffia, M.; Vergara, D. The multiverse nature of epithelial to mesenchymal transition. Semin Cancer Biol 2019, 58, 1-10, doi:10.1016/j.semcancer.2018.11.004.
- Stefania, D.; Vergara, D. The Many-Faced Program of Epithelial-Mesenchymal Transition: A System Biology-Based View. Front Oncol 2017, 7, 274, doi:10.3389/fonc.2017.00274.
Reviewer 2 Report
Dear authors and editor,
The manuscript titled ‘’ Molecular Network Profiling in Intestinal- and 2 Diffuse-Type Gastric Cancer ‘’ described the regulatory molecular networks altered in intestinal and diffuse-type of GC.
Authors shown that several microRNAs regulated the networks in intestinal- and diffuse-type GC. Additionally a section about EMT has been added. The manuscript is a significant step towards understanding the molecular mechanisms behind the development of gastric cancer subtypes and EMT pathway.
The manuscript does not differ much from the original version. However, it reads much better. In my opinion, the division into intestinal, diffuse is simple and may have clinical implications during surgical planning and treatment. By adding an analysis of the EMT pathway, the manuscript is more complicated.
In conclusion, the manuscript is scientifically good, although it needs to be simplified.
Thank you for your choice me as a reviewer.
Author Response
Thank you very much for your comments to further improve the manuscript. According to your comments, the manuscript has been simplified to have the division into intestinal and diffuse in the analysis of the EMT pathway.
<line 256-260> Table 8 has been simplified by the deletion of the column showing “Category of GOTERM_BP_DIRECT” and the addition of “Category of GOTERM_BP_DIRECT is listed.” in the legend as follows:
Table 8. Gene Ontology (GO) (Biological Process) of genes regulated in intestinal- and diffuse-type gastric cancer (GC). The total 2815 probe set IDs were analyzed for enrichment analysis in DAVID, which resulted in 2394 genes analyzed in GO Biological Process. Category of GOTERM_BP_DIRECT is listed.
<line 307-318> Figure 6 has been revised to have intestinal-(a) and diffuse-(b) type GC.
Figure 6. Canonical Pathways for Regulation of the EMT pathway in intestinal- and diffuse-type GC. Gene mapping and in silicoprediction of the upstream and downstream effects of activation or inhibition on molecules are shown in Canonical pathways for Regulation of the EMT pathway. The genes of which expression was altered in intestinal- and diffuse-type GC are shown in pink (up-regulated) or green (down-regulated). Predicted activation or inhibition is shown in orange or blue, respectively. (a) Gene expression and pathway activity prediction in intestinal-type GC are shown. (b) Gene expression and pathway activity prediction in diffuse-type GC are shown.
Reviewer 3 Report
The manuscript entitled “Molecular Network Profiling in Intestinal- and 2 Diffuse-Type Gastric Cancer” describes that the RefSeq data of intestinal- and diffuse-type GC has been analyzed for exploring the molecular networks and AI modeling application. Top 10 genes of which gene expression is altered in intestinal- and diffuse-type GC RefSeq data included CKS1B, CSE1L, DDX27, GET4, MRGBP, MSL3P1, PARD6B, RAE1, TOMM34 and YTHDF1. The network analysis of altered genes in intestinal- and diffuse-type GC generated networks related to cancer, gastrointestinal disease, organismal injury and abnormalities, amino acid metabolism, molecular transport, small molecule biochemistry, and so on. Several microRNAs including miR-205-5p, miR-21-5p, let-7a-5p, let-7, miR-24-3p, miR-291a-3p were identified to regulate networks involved in intestinal- and diffuse-type GC. Since previous studies have revealed the involvement of miR-200s in promoting metastatic colonization by inhibiting EMT and promoting mesenchymal-epithelial transition (MET), it may be a very interesting approach to reveal microRNA networks in EMT. The several microRNAs are involved and regulated in EMT and MET, which would be critical for progression and metastasis process. DataRobot Automated Machine Learning created prediction models to distinguish intestinal- and diffuse-type GC with results of up-stream analysis and the network picture data. The image recognition of molecular networks by AI would distinguish the intestinal- and diffuse-type GC.
Authors have well revised this manuscript. It is suggested to be accepted.
Author Response
Thank you very much for your comprehensive comments and suggestion to accept the manuscript. It is highly appreciated.
Round 2
Reviewer 1 Report
The paper has been improved and my comments addressed.
This manuscript is a resubmission of an earlier submission. The following is a list of the peer review reports and author responses from that submission.
Round 1
Reviewer 1 Report
In this manuscript, the authors used a bioinformatics approach to analyze TCGA data and to define molecular networks altered in intestinal- and diffuse-type gastric cancer (GC). Data were then analysed using a data robot automated machine learning approach. This allowed the creation of a prediction model to distinguish intestinal- and diffuse-type GC.
Functional data in support of these results are lacking thus limiting the translation implications. Moreover, I believe that is unclear the correlation with EMT.
Reviewer 2 Report
Dear authors and editor,
The manuscript titled ‘’ Molecular Network Profiling in Intestinal- and 2 Diffuse-Type Gastric Cancer ‘’ described the regulatory molecular networks altered in intestinal and diffuse-type of GC.
Authors shown that several microRNAs regulated the networks in intestinal- and diffuse-type GC. The manuscript is a significant step towards understanding the molecular mechanisms behind the development of gastric cancer subtypes. However, this attempt is a bit tricky for me. The manuscript is not systematized, it has many threads.
In conclusion, the manuscript needs some improvements.
Thank you for your choice me as a reviewer.
Reviewer 3 Report
Dear Editor,
The manuscript entitled “Molecular Network Profiling in Intestinal- and 2 Diffuse-Type Gastric Cancer” describes that the RefSeq data of intestinal- and diffuse-type GC has been analyzed for exploring the molecular networks and AI modeling application. Top 10 genes of which gene expression is altered in intestinal- and diffuse-type GC RefSeq data included CKS1B, CSE1L, DDX27, GET4, MRGBP, MSL3P1, PARD6B, RAE1, TOMM34 and YTHDF1. The network analysis of altered genes in intestinal- and diffuse-type GC generated networks related to cancer, gastrointestinal disease, organismal injury and abnormalities, amino acid metabolism, molecular transport, small molecule biochemistry, and so on. Several microRNAs including miR-205-5p, miR-21-5p, let-7a-5p, let-7, miR-24-3p, miR-291a-3p were identified to regulate networks involved in intestinal- and diffuse-type GC. Since previous studies have revealed the involvement of miR-200s in promoting metastatic colonization by inhibiting EMT and promoting mesenchymal-epithelial transition (MET), it may be a very interesting approach to reveal microRNA networks in EMT. The several microRNAs are involved and regulated in EMT and MET, which would be critical for progression and metastasis process. DataRobot Automated Machine Learning created prediction models to distinguish intestinal- and diffuse-type GC with results of up-stream analysis and the network picture data. The image recognition of molecular networks by AI would distinguish the intestinal- and diffuse-type GC.
- Authors should introduce the difficulty in how to distinguish the diffuse-type gastric GC with intestinal-type GC until now; the importance in distinguishing the two type GC.
- Authors should introduce how important and potential to use the molecular network profile to distinguish diffuse- and intestinal- type GC.
- Is it easier to distinguish the diffuse- and intestinal- type GC using this molecular network profile than traditional/previous approaches?
- Is this model “ Molecular Network Profiling” easy to be used and applied in clinical medicine? For example, detecting these Top 10 genes/proteins, and miR-205-5p, miR-21-5p, let-7a-5p, let-7, miR-24-3p, miR-291a-3p from blood. It is always more complexed to analyze genes, proteins and miRNAs et al from tissues than blood.